# Insecurity of Quantum Blockchains Based on Entanglement in Time

**DOI:** 10.3390/e25091344

**Published:** 2023-09-16

**Authors:** Piotr Zawadzki

**Affiliations:** Department of Telecommunications and Teleinformatics, Silesian University of Technology, Akademicka 16, 44-100 Gliwice, Poland; piotr.zawadzki@polsl.pl

**Keywords:** quantum cryptography, quantum entanglement, quantum blockchain

## Abstract

In this study, the security implications of utilizing the concept of entanglement in time in the quantum representation of a blockchain data structure are investigated. The analysis reveals that the fundamental idea underlying this representation relies on an uncertain interpretation of experimental results. A different perspective is provided by adopting the Copenhagen interpretation, which explains the observed correlations in the experiment without invoking the concept of entanglement in time. According to this interpretation, the qubits responsible for these correlations are not entangled, posing a challenge to the security foundation of the data structure. The study incorporates theoretical analysis, numerical simulations, and experiments using real quantum hardware. By employing a dedicated circuit for detecting genuine entanglement, the existence of entanglement in the process of generating a quantum blockchain is conclusively excluded.

## 1. Introduction

The term blockchain is interchangeably used to refer to two related but distinct notions: the blockchain data structure and the blockchain technology. The former denotes a specific way of data organization and storage. In this meaning, the blockchain is a ledger that records a series of events in a chronological and immutable manner. The data structure is composed of blocks that are cryptographically linked in a chain-like structure. Each block includes a reference to the previous block, forming a secure and transparent sequence of data. The primary purpose of the blockchain data structure is to ensure the integrity and immutability of the recorded data. Blockchain technology is a broader concept that encompasses various tools, protocols, and algorithms that enable the functioning of a distributed ledger built from a set of blockchain data structures. The decentralized management enables high BFT and provides the foundational infrastructure for creating trust, facilitating peer-to-peer transactions, and enabling decentralized applications to operate without a central authority.

The concept of data chaining dates to 1990 [1,2], when it was proposed in the context of timestamping documents and provision of tamper-proof logs. The immutability of blockchain, i.e., impossibility to alter the existing elements of a chain without changing all subsequent entries, is closely related to the properties of one-way hash functions. Hash functions take an input of arbitrary length and produce a fixed-size output, known as a hash or digest. One-way hash functions permit easy calculation of output, but finding the input, i.e., preimage, that leads to a specific output is computationally infeasible. Each blockchain’s block contains a cryptographic hash that is calculated based on the contents of the block, including the hash of the previous block. If any data in a block are tampered with, the hash of that block changes, breaking the chain’s continuity and indicating manipulation. It follows that properties of one-way hash functions play a crucial role in the security and integrity of a blockchain data structure.

The Bitcoin Whitepaper [3] is generally considered the pivotal document. It not only defined the blockchain data structure but also proposed a consensus protocol with its killer application—a decentralized cryptocurrency. The ingenious combination of these two ingredients establishes the basis for the development of cryptocurrencies and the wider adoption of blockchain technology. Beyond cryptocurrencies, blockchain technology is being explored for various applications, including supply chain management, voting systems, healthcare records, intellectual property protection, and more, where the properties of immutability provide significant benefits. Presently, there exist many consensus protocols that manage the distributed ledger in different ways, but the cryptography plays a crucial role in all implementations. Techniques like digital signatures, hashing algorithms, and cryptographic proofs are used to verify the authenticity and integrity of transactions and blocks.

The security of all these elements, and, in consequence, the blockchain technology as a whole, are affected in some way by advances in quantum computing. Although specific quantum attacks against one-way hash functions are in their infancy [4,5,6,7,8], it is known that the Grover search algorithm places quantum-enabled participants of the systems in a privileged position. They can potentially mine more efficiently in incentive-based protocols, or it is easier for them to change the past, i.e., falsify blocks stored in a blockchain, due to better performance of a brute force preimage attack. However, the most devastating is Shor’s algorithm as it potentially invalidates all asymmetric cryptography [9,10] and potentially some symmetric primitives [11,12] used in the consensus protocol.

The construction of post-quantum classical hash functions and digital signatures that are resistant to known quantum attacks [7,8,13] is one of the possible ways out of this difficult situation. A holistic proposal of a blockchain system based on the above principles is presented in [14]. The second approach is based on the assumption that cryptographic tools built upon quantum properties of the matter can be resistant to quantum attacks. Research on this subject is in its infancy; different quantum information processing techniques are applied and different aspects of security are addressed.
Jogenfors in [15] proposed a quantum data structure and protocols that emulate the behavior of Bitcoin. The security of this solution is based on the no-cloning principle.The work of Kiktenko et al. [16] addresses the security of consensus. The proposed protocol authenticates messages with symmetric keys. Unconditional security is accomplished with a separate QKD network that is responsible for provisioning users with OTP keys.Wang et al. [17] combined the classical consensus algorithm DPoSB [18] with quantum signature based on quantum state computational distinction with a fully flipped permutations problem [19]. The representation of blockchain data is purely classical, although the used algorithm eliminated the need of hash function use.On the other hand, the work of Rajan et al. [20] focuses on the creation of a quantum data structure that can be used for immutable data storage. The proposed design is founded on a phenomenon called entanglement in time.Gao et al. in [21] continued that concept and supported the quantum blockchain with a consensus protocol following the DPoS paradigm.

In our study, we examine the security implications of the quantum representation employed in the blockchain data structure, as discussed in previous works [20,21]. Our analysis demonstrates that the core concept underlying this representation relies on an uncertain interpretation of the experimental findings presented in [22]. By offering an alternative viewpoint rooted in the Copenhagen interpretation, we provide an explanation for the observed correlations without invoking the concept of entanglement in time. According to our interpretation, the qubits contributing to these correlations are not entangled, thus challenging the security basis of the data structure, which depends on the notion of entanglement in time.

It is imperative to emphasize that, considering the presented results, there currently exists no quantum analogue of the blockchain ledger. To date, the scope of research has predominantly encompassed quantum-enabled adaptations of consensus protocols. This implies that, while considerable efforts have been directed towards fortifying consensus algorithms against quantum threats, a comprehensive quantum representation of the entire blockchain ledger has yet to materialize. Researchers have primarily concentrated on the mitigation of potential quantum vulnerabilities within blockchain technology by augmenting consensus mechanisms rather than embarking on the transformation of the underlying ledger structure into the quantum domain. The post-quantum proposals, which hinge on the assumption that cryptographic algorithms invulnerable to efficient quantum attacks today will remain secure in the future, lack a solid mathematical foundation. Consequently, it becomes evident that the development of quantum-resistant data structures holds paramount importance in sustaining the long-term integrity of data preserved on blockchain networks. Such solutions should be poised for implementation when technology matures to a stage where their deployment becomes feasible.

## 2. Quantum Blockchain

The concept of a quantum blockchain, which aims to preserve the sequential ordering of events, is founded on experimental findings presented in works [22,23]. The former work introduces a modified approach, based on the procedure described in [24], that allows for the accumulation of entanglement from non-concurrently existing EPR pairs within a GHZ state stored in quantum memory. This modification enables the accumulation of entanglement from EPR pairs existing at different temporal points. The latter work focuses on a modification to the entanglement swapping procedure [25,26]. Through this modification, researchers demonstrate the existence of correlations similar to the ones existing in entanglement swapping procedure but for pairs of photons that do not coexist in time. The interpretation of measurement results presented in [22] suggests the intriguing possibility of generating EPR pairs from photons that do not exist simultaneously. This interpretation characterizes the phenomenon as “entanglement in time”, highlighting the temporal nature of entangled state formation. The potential implications of this result for the development of a quantum blockchain have been outlined in [20]. The presented further process of creating a quantum blockchain is a nearly direct representation of the expressions and arguments given in that work.

Let us consider the process depicted in Figure 1. At time t=0, the EPR pair ${\left|\beta_{1,1}\rangle \right.}_{\mathrm{AB}}$ is created, while, at time t=τ, the pair ${\left|\beta_{1,1}\rangle \right.}_{\mathrm{CD}}$ is formed. Adapting the description used in entanglement swapping to the presented situation, the state of the system is described by expression
(1)$${\left|\beta_{1,1}\rangle \right.}_{A,B}^{0,0}⊗{\left|\beta_{1,1}\rangle \right.}_{C,D}^{\tau ,\tau }\;,$$
where ${\left|\beta_{m,n}\rangle \right.}_{X,Y}^{t_{1},t_{2}}=\frac{1}{\sqrt{2}}\left({\left|0\rangle \right.}_{X}^{t_{1}}{\left|n\rangle \right.}_{Y}^{t_{2}}+{\left(-1\right)}^{m}{\left|1\rangle \right.}_{X}^{t_{1}}{\left|n⊕1\rangle \right.}_{Y}^{t_{2}}\right)$. In the above expression, the superscript indicates time. Subsequently, photons B and D are delayed by a time τ, resulting in the representation of the system state
(2)$$\begin{array}{c} {\left|\beta_{1,1}\rangle \right.}_{A,B}^{0,\tau }⊗{\left|\beta_{1,1}\rangle \right.}_{C,D}^{\tau ,2\tau }=\frac{1}{2}\left(-{\left|\beta_{0,0}\rangle \right.}_{A,D}^{0,2\tau }{\left|\beta_{0,0}\rangle \right.}_{B,C}^{\tau ,\tau }+{\left|\beta_{0,1}\rangle \right.}_{A,D}^{0,2\tau }{\left|\beta_{0,1}\rangle \right.}_{B,C}^{\tau ,\tau }\right.+\\ \;\left.+{\left|\beta_{1,0}\rangle \right.}_{A,D}^{0,2\tau }{\left|\beta_{1,0}\rangle \right.}_{B,C}^{\tau ,\tau }-{\left|\beta_{1,1}\rangle \right.}_{A,D}^{0,2\tau }{\left|\beta_{1,1}\rangle \right.}_{B,C}^{\tau ,\tau }\right). \end{array}$$
Photons BC measured at time slot t=τ result in the collapse of the state of the remaining photons into one of the possible EPR pairs that entangles photon A at time t=0 with photon D at time t=2τ. The experiment demonstrated in [22] revealed that photons A,D are correlated in a way determined by the outcome of measurement BC. To be more specific, the post-selection on ${\left|\beta_{m0}\rangle \right.}_{\mathrm{BC}}$ has been used and equality a=d observed.

Two or more EPR pairs can be merged into a GHZ state using the fusion process proposed in [23]. For instance, merging two EPR pairs into a GHZ state composed of 4 photons requires delay and PBS ([20], Equation (8))
(3)$$\left|GHZ_{n_{0},n_{1},n_{2},n_{3}}^{0,\tau ,\tau ,2\tau }\rangle \right.=\frac{1}{\sqrt{2}}\left({\left|0\rangle \right.}^{0}{\left|n_{1}\rangle \right.}^{\tau }{\left|n_{2}\rangle \right.}^{\tau }{\left|n_{3}\rangle \right.}^{2\tau }+{\left(-1\right)}^{n_{0}}{\left|1\rangle \right.}^{0}{\left|{\bar{n}}_{1}\rangle \right.}^{\tau }{\left|{\bar{n}}_{2}\rangle \right.}^{\tau }{\left|{\bar{n}}_{3}\rangle \right.}^{2\tau }\right)\;.$$
The process proposed in [23] is extendible and permits fusing additional photons at later times
(4)$$\begin{array}{cc} \left|GHZ_{n_{0},n_{1},n_{2},\ldots ,n_{2k-3},n_{2k-2},n_{2k-1}}^{0,\tau ,\tau ,\ldots ,(k-1)\tau ,(k-1)\tau ,k\tau }\rangle \right.= & \frac{1}{\sqrt{2}}\left(\left|0^{0}n_{1}^{\tau }n_{2}^{\tau }\ldots n_{2k-3}^{(k-1)\tau }n_{2k-2}^{(k-1)\tau }n_{2k-1}^{k\tau }\rangle \right.+\right.\\ & \left.+{\left(-1\right)}^{n_{0}}\left|1^{0},{\bar{n}}_{1}^{\tau },{\bar{n}}_{2}^{\tau },\ldots ,{\bar{n}}_{2k-3}^{(k-1)\tau },{\bar{n}}_{2k-2}^{(k-1)\tau },{\bar{n}}_{2k-1}^{k\tau }\rangle \right.\right)\;. \end{array}$$
The consensus protocol proposed in [20] verifies the entanglement of the state described in (Equation 4) using the procedure described in [27].

The security of the blockchain data structure designed that way is rooted in the ability to generate states as depicted in Equation (Equation 4). These states exhibit entanglement between photons that do not coexist concurrently, revealing non-classical correlations during measurements. As emphasized by the designers, the process of encoding information within these states establishes a profound connection, not merely with a historical record but with the authentic state of the system at a specific moment.

In our subsequent analysis, we will demonstrate that the fundamental correlations underlying such a construction, as observed in experiment [22], can be explained within the framework of the “conventional” Copenhagen interpretation of quantum mechanics. This explanation does not rely on the concept of temporal entanglement but rather embraces the principle of causality. Furthermore, it is noteworthy that the observation of these correlations does not require photons A and D to remain entangled, suggesting that capturing the historical record in the form of state (Equation 4) is infeasible.

## 3. Analysis

Let us consider again the scheme from Figure 2 but this time from the viewpoint of Copenhagen interpretation. This interpretation assumes that measurement may have changed the state of the measured object. In consequence, the order in which measurements and transformations are applied is important. Figure 2 presents a scheme of entangled in time EPR pair creation with marked time slices. Let us analyze the system state in these moments of time.
Slice 0. The AB pair is created
$$\left|{\mathrm{slice}}_{0}\rangle \right.={\left|\beta_{11}\rangle \right.}_{\mathrm{AB}}=2^{-1/2}\left({\left|0\rangle \right.}_{A}{\left|1\rangle \right.}_{B}-{\left|1\rangle \right.}_{A}{\left|0\rangle \right.}_{B}\right).$$Slice 1. The qubit A is measured. The output *a* is random and $p_{A}a=1/2$
$$\left|{\mathrm{slice}}_{1}\left(a\right)\rangle \right.={\left|a\rangle \right.}_{A}{\left|\bar{a}\rangle \right.}_{B}\;.$$Slice 2. The CD pair is created
$$\left|{\mathrm{slice}}_{2}\left(a\right)\rangle \right.=\left|{\mathrm{slice}}_{1}\left(a\right)\rangle \right.{\left|\beta_{11}\rangle \right.}_{\mathrm{CD}}=\frac{1}{\sqrt{2}}{\left|a\rangle \right.}_{A}{\left|\bar{a}\rangle \right.}_{B}\left({\left|0\rangle \right.}_{C}{\left|1\rangle \right.}_{D}-{\left|1\rangle \right.}_{C}{\left|0\rangle \right.}_{D}\right)\;.$$Slice 3. The first step of Bell measurement—application of ${CX}_{B\to C}$ gate
$$\begin{array}{cc} \left|{\mathrm{slice}}_{3}\left(a\right)\rangle \right. & ={CX}_{\mathrm{BC}}\left|{\mathrm{slice}}_{2}\left(a\right)\rangle \right.=\;\\ & =\frac{1}{\sqrt{2}}{\left|a\rangle \right.}_{A}\left({CX}_{\mathrm{BC}}{\left|\bar{a}\rangle \right.}_{B}{\left|0\rangle \right.}_{C}{\left|1\rangle \right.}_{D}-{CX}_{\mathrm{BC}}{\left|\bar{a}\rangle \right.}_{B}{\left|1\rangle \right.}_{C}{\left|0\rangle \right.}_{D}\right)=\\ & =\frac{1}{\sqrt{2}}{\left|a\rangle \right.}_{A}{\left|\bar{a}\rangle \right.}_{B}\left({\left|\bar{a}\rangle \right.}_{C}{\left|1\rangle \right.}_{D}-{\left|a\rangle \right.}_{C}{\left|0\rangle \right.}_{D}\right). \end{array}$$Slice 4. Application of the Hadamard gate
(5)$$\begin{array}{cc} \left|{\mathrm{slice}}_{4}\left(a\right)\rangle \right. & =H_{B}\left|{\mathrm{slice}}_{3}\left(a\right)\rangle \right.=\\ & =\frac{1}{2}{\left|a\rangle \right.}_{A}\left({\left|0\rangle \right.}_{B}+{\left(-1\right)}^{\bar{a}}{\left|1\rangle \right.}_{B}\right)\left({\left|\bar{a}\rangle \right.}_{C}{\left|1\rangle \right.}_{D}-{\left|a\rangle \right.}_{C}{\left|0\rangle \right.}_{D}\right)=\;\\ & =\left\{\begin{array}{cc} \frac{1}{2}{\left|0\rangle \right.}_{A}{\left|+\rangle \right.}_{B}\left({\left|1\rangle \right.}_{C}{\left|1\rangle \right.}_{D}-{\left|0\rangle \right.}_{C}{\left|0\rangle \right.}_{D}\right) & a=0\;,\\ \frac{1}{2}{\left|1\rangle \right.}_{A}{\left|-\rangle \right.}_{B}\left({\left|0\rangle \right.}_{C}{\left|1\rangle \right.}_{D}-{\left|1\rangle \right.}_{C}{\left|0\rangle \right.}_{D}\right) & a=1\;. \end{array}\right. \end{array}$$
Table 1 presents a comprehensive summary of the recorded results obtained from registers A, B, and C, alongside their corresponding states after measurement. It becomes apparent that the value of *c* plays a unique role in determining whether correlation or anti-correlation manifests between registers A and D. Specifically, when the value of *c* equals 0, correlation is observed. Consequently, when post-selecting the recorded outcomes for c=0 or c=1, it induces correlation or anti-correlation in the measurement results. Therefore, the observed correlations align with the findings documented in the experiment described in [22].

More phenomenological explanation of these observations comes from the fact that the procedure initialized at time t=τ is just a qubit teleportation from register B to D with state correction in target register removed. The measurement of qubit A of the EPR pair ${\left|\beta_{1,1}\rangle \right.}_{A,B}$ at time t=0 induces a post-measurement state ${\left|\bar{a}\rangle \right.}_{B}$ in register B. Its teleportation, without correction, causes register D to be in state $X^{c}Z^{b}{\left|\bar{a}\rangle \right.}_{D}$. Making post-selection on *c* value is equivalent to the selection of bit-flip correction, which is, interestingly, the same behavior one would observe in dual basis. This comes from the fact that roles of **Z** and **X** in dual basis are exchanged: **Z** becomes bit-flip operation and **X** phase-flip. As a consequence, the post-selection on value of *b* of A and D measurements in dual basis will select their correlation or anti-correlation.

The calculations provided above offer an explanation for the observed correlations described in [22]. However, a thorough analysis of the summarized post-measurement states presented in Table 1 reveals that qubits A and D were never entangled. The observed correlation between their measurement outcomes can be solely attributed to the principle of causality: the measurement result of qubit D is entirely determined by the previously observed values of qubits A and C. These findings stand in contrast to the outcomes derived from the analysis of a basic entanglement swapping circuit depicted in Figure 3, where the final state is represented by Equation (Equation 6).
$$\begin{array}{c} \left|{\mathrm{slice}}_{2}\rangle \right.=\frac{1}{2}\left(-{\left|\beta_{0,0}\rangle \right.}_{A,D}{\left|\beta_{0,0}\rangle \right.}_{B,C}+{\left|\beta_{0,1}\rangle \right.}_{A,D}{\left|\beta_{0,1}\rangle \right.}_{B,C}\right.+\;\\ \;+\left.{\left|\beta_{1,0}\rangle \right.}_{A,D}{\left|\beta_{1,0}\rangle \right.}_{B,C}-{\left|\beta_{1,1}\rangle \right.}_{A,D}{\left|\beta_{1,1}\rangle \right.}_{B,C}\right). \end{array}$$
(6)$$\left|{\mathrm{slice}}_{5}\rangle \right.={\left|b\rangle \right.}_{B}{\left|c\rangle \right.}_{C}{\left|\beta_{b,c}\rangle \right.}_{A,D}\;.$$

## 4. Cloud-Based Quantum Experiment

Theoretical considerations can be verified in classical simulators and real quantum hardware. The quantum computing experiments presented in this study [28] were conducted using *IBM Quantum Lab*, an online platform that provides access to IBM Quantum systems. We acknowledge the use of IBM Quantum system ibmq_lima available through *IBM Quantum Lab*. The simulations were performed using Qiskit (version 0.43.2) and the IBM Quantum software stack. The Jupyter Notebooks used for simulation and interfacing with quantum computers are available as supplemental material to this contribution.

Given the probabilistic nature of quantum computers, it is imperative to execute numerous iterations of the quantum circuits to extract the probability distribution governing the observed outcomes. In our computational experiment, each quantum circuit has undergone simulation for 4000 iterations. The outcomes registered in quantum registers B and C play a pivotal role in determining the nature of correlation manifested in quantum registers A and D. Given the existence of four distinct BC combinations, each post-selected series comprises approximately 1000 recorded measurements. The observed coincidence of values in registers A and D is subject to the following interpretation: a coincidence of 100% signifies a state of perfect correlation, a coincidence of 0% signifies complete anti-correlation, while a coincidence of 50% implies the absence of any correlation.

We have conducted simulations and quantum computations to validate the expected correlation between measurements in computational and dual bases for the circuits depicted in Figure 2 and Figure 3. The analysis of the obtained results, as summarized in Table 2 and Table 3, demonstrates strong agreement between the simulation and outcomes from real hardware. However, it is important to note that the presented results do not provide conclusive evidence regarding the presence or absence of entanglement in the generation of entanglement in time.

Confirming the presence of entanglement through experimental verification poses a significant challenge. Merely observing coincidences in measurements performed in mutually unbiased bases is insufficient to establish the existence of entanglement, as explained in the preceding paragraph. A more comprehensive approach involves employing a Bell measurement circuit to detect genuine entanglement. This circuit produces deterministic outcomes when a specific type of EPR pair is measured while yielding stochastic outcomes otherwise. Consequently, it becomes feasible using available quantum computers to measure the states occurring in registers AD on the output of circuits depicted in Figure 2 and Figure 3 using this approach. The values of parameters *b* and *c* provide information about the purported type of the measured EPR pair. By performing post-selection based on these values, it becomes possible to select cycles that involve a specific EPR pair type and verify whether the outcomes of the Bell measurement circuit are deterministic or stochastic. The circuits shown in Figure 4 were both simulated and executed on real quantum hardware. In the case of entanglement swapping, the classical simulation exhibited 100% coincidence with the expected outputs, while the execution on actual hardware yielded 71% agreement. Similarly, for the entanglement in time generation circuit, the corresponding values were 49% for classical simulation and 41% for execution. Considering that coincidence around 50% indicates no correlation, the presented computational experiment effectively excludes the presence of entanglement in this scenario.

## 5. Conclusions

This study encompassed theoretical analysis, numerical simulations, and experiments involving real quantum hardware to investigate various aspects of entanglement. The research outcomes are as follows:*Correlation of measurement outcomes in mutually unbiased bases is not sufficient proof of entanglement*. It was determined that the correlation observed in measurement outcomes, particularly in mutually unbiased bases, does not provide conclusive evidence of entanglement. Additional tests and criteria are necessary to establish the presence of genuine entanglement.*Absence of entanglement in the generator of entanglement in time*. The results demonstrated the absence of genuine entanglement in this specific system. The combination of analysis and experimental validation allowed for a comprehensive understanding of the correlations observed, indicating the lack of entanglement.*Correlation observed for entanglement in time generator can be explained within the framework of the Copenhagen interpretation of quantum mechanics*. The research established that the observed correlation arising in the generator of the entanglement in time can be fully explained within the framework of the Copenhagen interpretation of quantum mechanics. The analysis took into account the probabilistic nature of quantum phenomena and the fundamental role of measurement.

Blockchains are architecturally engineered to provide robust security and immutability over extended temporal horizons, frequently spanning decades or even centuries. Their applications extend beyond the realm of cryptocurrencies, encompassing diverse domains, including the documentation of legal contracts, registration of land ownership, tracking of supply chain data, and management of healthcare records. Ensuring the indisputable integrity of this multifaceted data repository assumes paramount significance. The advent of quantum computing poses a substantial threat to the preservation of this integrity. The development of quantum-resistant data structures emerges as an imperative strategy for upholding the trustworthiness of information archived within blockchain systems.

These research outcomes contribute to our understanding of entanglement and its use as a resource for building a quantum distributed ledger. The presented findings emphasize the importance of adhering to the Occam’s Razor principle when interpreting experimental observations. Introducing novel entities or concepts, such as entanglement in time, is unnecessary if existing principles and phenomena can account for the observed results. Verification of the existence of entanglement based solely on the presence of unusually high correlations is insufficient if causality cannot be excluded, as is the case with the entanglement in time generator. Additionally, the study demonstrates that the entanglement in time cannot be utilized as a tool to construct quantum data structures capable of storing an immutable history of events. As a result, proposals for quantum blockchain systems relying on this property inherently lack security by design.

These conclusions highlight the need for careful consideration of the underlying principles and limitations when investigating entanglement phenomena. They contribute to the ongoing scientific discourse surrounding the feasibility and interpretation of entanglement-related concepts, providing insights into the security implications and practical applications of quantum information processing systems.

The principle articulated by Nicolaus Copernicus, stating that “bad money drives out good” finds relevance in the scientific landscape. The prevailing practice of prioritizing the dissemination of new findings, coupled with the difficulty in publishing verification studies, can contribute to the circulation of unconfirmed results within the scientific community. Consequently, these weakly verified findings may serve as the basis for subsequent layers of systems or research, with unsuspecting readers implicitly assuming the correctness of such published contributions. Complicating matters further, the research funding system, often based on grants and subject to market dynamics, can introduce influences reminiscent of Copernicus’ law. The above observations emphasize the importance of addressing these challenges in the scientific community. By fostering a culture that encourages rigorous verification, replication, and critical evaluation of findings, we can guard against the risks posed by unconfirmed or misleading results.

## Figures and Tables

**Figure 1 entropy-25-01344-f001:**
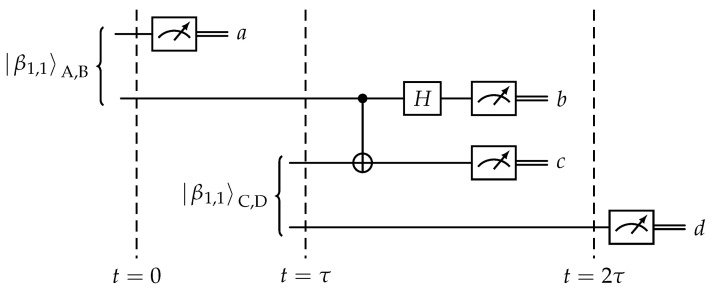
Entanglement of qubits that never coexisted.

**Figure 2 entropy-25-01344-f002:**
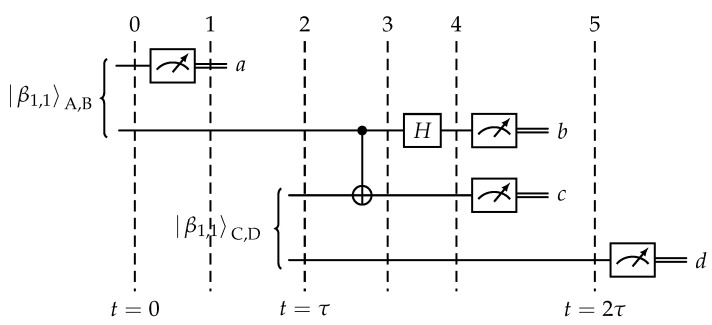
Generation of an EPR pair entangled in time.

**Figure 3 entropy-25-01344-f003:**
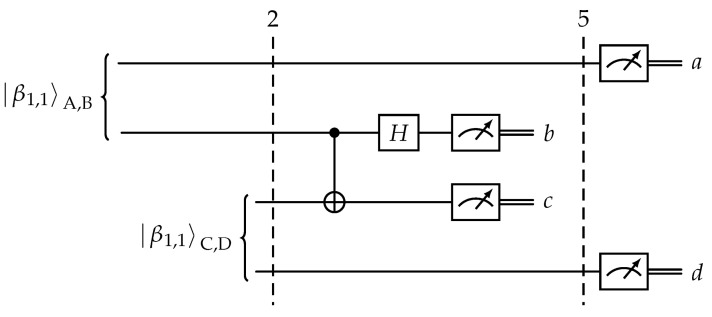
Entanglement swapping circuit.

**Figure 4 entropy-25-01344-f004:**
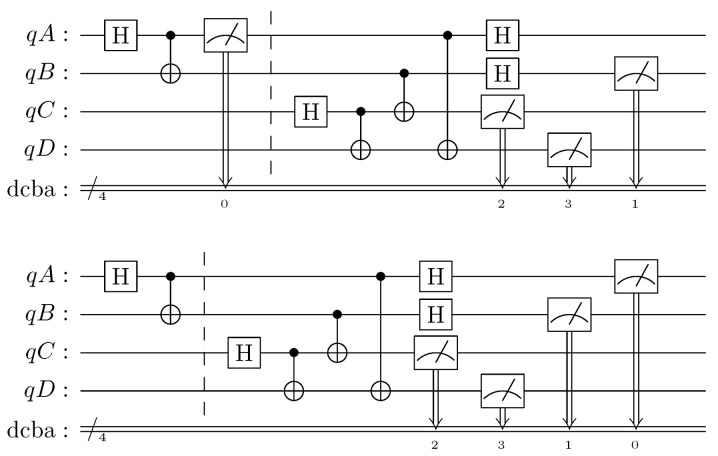
Circuits used in genuine entanglement detection: generator of entanglement in time (**top**) and entanglement swapping (**bottom**).

**Table 1 entropy-25-01344-t001:** Outcomes observed on registers A, B, and C along with the corresponding post-measurement states.

*a*	*c*	*b*	Post-Measurement State	*a*	*c*	*b*	Post-Measurement State
0	0	0	${\left|0\rangle \right.}_{A}{\left|0\rangle \right.}_{B}{\left|0\rangle \right.}_{C}{\left|0\rangle \right.}_{D}$	0	0	1	${\left|0\rangle \right.}_{A}{\left|1\rangle \right.}_{B}{\left|0\rangle \right.}_{C}{\left|0\rangle \right.}_{D}$
0	1	0	${\left|0\rangle \right.}_{A}{\left|0\rangle \right.}_{B}{\left|1\rangle \right.}_{C}{\left|1\rangle \right.}_{D}$	0	1	1	${\left|0\rangle \right.}_{A}{\left|1\rangle \right.}_{B}{\left|1\rangle \right.}_{C}{\left|1\rangle \right.}_{D}$
1	0	0	${\left|1\rangle \right.}_{A}{\left|0\rangle \right.}_{B}{\left|0\rangle \right.}_{C}{\left|1\rangle \right.}_{D}$	1	0	1	${\left|1\rangle \right.}_{A}{\left|1\rangle \right.}_{B}{\left|0\rangle \right.}_{C}{\left|1\rangle \right.}_{D}$
1	1	0	${\left|1\rangle \right.}_{A}{\left|0\rangle \right.}_{B}{\left|1\rangle \right.}_{C}{\left|0\rangle \right.}_{D}$	1	1	1	${\left|1\rangle \right.}_{A}{\left|1\rangle \right.}_{B}{\left|1\rangle \right.}_{C}{\left|0\rangle \right.}_{D}$

**Table 2 entropy-25-01344-t002:** Coincidence of measurements in computational basis. Coincidence values equal to 100%, 0%, and 50% are signs of perfect correlation, perfect anti-correlation, and no correlation, respectively.

Post-Selection		Entanglement in Time		Entanglement Swapping	
b	c	**Simulation [%]**	**Execution [%]**	**Simulation [%]**	**Execution [%]**
0	0	100.0	91.07	100.0	92.39
0	1	0.0	7.04	0.0	5.52
1	0	100.0	92.12	100.0	91.20
1	1	0.0	5.98	0.0	6.95

**Table 3 entropy-25-01344-t003:** Coincidence of measurements in dual basis. Coincidence values equal to 100%, 0%, and 50% are signs of perfect correlation, perfect anti-correlation, and no correlation, respectively.

Post-Selection		Entanglement in Time		Entanglement Swapping	
b	c	**Simulation [%]**	**Execution [%]**	**Simulation [%]**	**Execution [%]**
0	0	100.0	69.54	100.0	86.08
0	1	100.0	67.61	100.0	87.89
1	0	0.0	31.00	0.0	14.69
1	1	0.0	30.55	0.0	10.52

## Data Availability

This contribution is accompanied by Jupyter Notebooks that permit verification of the presented theses. They are available free of charge and are intended for non-commercial use and licensed under the Creative Commons Attribution Non-Commercial 4.0 International License. The authors and contributors of the software shall not be held liable for any damages, loss of data, or any other consequences resulting from the use or misuse of the software. The software includes third-party libraries or components, which are subject to their respective licenses and terms.

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
