# Peer review of "Insecurity of Quantum Blockchains Based on Entanglement in Time"

_entropy, 2023, doi:10.3390/e25091344_

Round 1
Reviewer 1 Report
As a reviewer of this paper, I have carefully read through the manuscript and would like to provide my feedback on the following questions:
1. The introduction of the paper provides a clear and concise overview of the research topic and its significance. The authors have provided sufficient background information and relevant references to support their research. However, I suggest that the authors further elaborate on the research gap and research questions to provide a more comprehensive understanding of the study.
2. The authors have cited a range of relevant references to support their research. However, I suggest that the authors consider including more recent studies to ensure that the paper is up-to-date with the latest research in the field.
3. The research design is appropriate for the research question and objectives. The authors clearly explain the research design and methodology, which is appropriate for the research question. However, I suggest the authors increase the comparison with other relevant papers to demonstrate their research contribution and uniqueness better. This can help readers better understand the importance and contribution of this paper and distinguish it from other related studies.
4. The methods used in the study are adequately described, and the authors have provided a clear explanation of the data collection and analysis methods. However, the authors should provide a more in-depth comparison and analysis of other relevant studies in the discussion and conclusion sections to better showcase their research results and contributions.
5. The study's results are presented, and the authors have used appropriate tables and figures to present the data. However, I suggest the authors consider including more detailed explanations of the statistical tests used to analyze the data.
6. The study's results support the conclusions drawn by the authors. The authors have provided a clear explanation of the study's implications and highlighted the research's limitations. However, I suggest the authors consider including more recommendations for future research to provide a more comprehensive understanding of the topic.
Author Response
Dear Reviewer,
We would like to express our gratitude for your valuable comments and feedback. The raised concerns are addressed in points that correspond to the formatting of the review.
1. Thank you for the suggestion. The introduction has been supplemented with two paragraphs that, in my opinion, remove pointed out deficiency.
2. This paper primarily focuses on the quantum representation of the blockchain ledger, a topic that, while crucial, has not gained widespread attention, possibly due to the inherent challenges associated with obtaining affirmative results in this domain. Furthermore, it is noteworthy that a majority of publications in the field of quantum blockchain have predominantly centered around quantum consensus protocols, with limited exploration of quantumizing the blockchain data itself. We have conscientiously considered the seminal works in this field and have ensured that the pertinent literature has been adequately referenced. It is important to underscore that the core contribution of this paper revolves around a reinterpretation of the findings presented in reference [22]. An analysis of citations associated with the aforementioned reference reveals that it has garnered limited attention in the field of quantum cryptography, with only one publication within the realm of quantum blockchain making reference to it. We acknowledge that the implications of our work may extend beyond the scope presented in this paper; however, we believe that a comprehensive examination of these broader ramifications deserves a dedicated investigation in future research.
3. I'd like to extend my appreciation for bringing up this pertinent query.
As of the current state of research, only two proposals have delved into the exploration of temporal entanglement for the construction of a quantum ledger in the context of blockchain technology. These seminal works are duly cited within submitted manuscript, specifically as references [20] and [21]. The methodology by which these proposals introduce the concept of a quantum ledger is comprehensively elucidated in Section 2 of our paper, aptly titled "Quantum Blockchain." It is worth noting that their approach essentially builds upon and refines the formalism initially presented in reference [22], albeit employing a more elaborate and sophisticated framework.
4. The outcomes of the experiment documented in reference [22] are elucidated through the concept of temporal entanglement. This particular formalism has been employed in the development of quantum blockchain ledgers as demonstrated in references [19] and [20]. The principal focus of my contribution is to illustrate that the findings from reference [22] can be comprehensively explained within the framework of standard quantum formalism and the principle of causality and without the need for the introduction of ill defined constructs. To the best of my knowledge, there are no alternative methods available to explain the correlations observed in reference [22].
5. I appreciate your diligence in highlighting this concern. In response, I have incorporated a paragraph in section "Experiment" that elucidates the methodology employed for generating the calculations in question, as well as providing clear guidance on how to interpret the data presented in the tables.
6. Thank you for pointing that out. The adequate passage has been added to the conclusion.
I've included the updated manuscript with highlighted changes I made in response to your comments along with this message. Once again, I sincerely appreciate your insightful feedback and I'm committed to addressing any further queries or concerns to enhance the quality and rigor of my work.
Best regards,
Piotr ZAWADZKI

Reviewer 2 Report
This is an important paper that re-analyzes recent proposals and experiments that aim to test the possibility of entanglement in time. As the paper clearly demonstrates, identical outcomes can be derived if one does not assume entanglement in time but instead keeps track of causal relationships between quantum states. The analysis has implications not only for a conceptual understanding of quantum mechanics, but also, potentially, for the security of certain quantum protocols.
In general the writing is good, but there are minor inaccuracies for instance in the application of articles.
Author Response
Dear Reviewer,
Thank you for your positive feedback. I'm grateful for your favorable assessment of my work.
Best regards,
Piotr ZAWADZKI
Reviewer 3 Report
The author discusses the role of quantum entanglement in the quantum blockchain and conducts verification experiments. The work examines the presence or absence of entanglement due to the timing of measurements in circuits, such as sequential quantum teleportation and entanglement swapping, using simple mathematical formulas. I feel that it only describes the basic principles and lacks adequate details on the security of the quantum blockchain protocol, but I think it is worthwhile as a verification experiment for entanglement with IBMQ. Overall, this manuscript is clearly written and the technical content is understandable. Thus, I recommend the publication of this manuscript.
Author Response
Dear Reviewer,
I genuinely appreciate your positive feedback on my work. I intentionally left out the security analysis of consensus protocols in quantum ledger systems that rely on temporal entanglement. I did so because I believed it would be confusing for readers and not a good use of my time. These protocols depend on creating entanglement over different "snapshots of history," as shown in equation 4. My main point in this contribution is to demonstrate that it's not possible to create such entangled pairs as described in [22]. So, if we can't establish the foundational data structure with the desired properties, there's no point in analyzing protocols built on top of it.
Best regards,
Piotr ZAWADZKI